# DLLMQuant: A Post-Training Quantization Framework Tailored for Diffusion-Based Large Language Models

## Abstract

Diffusion-based large language models (DLLMs) have shown promise for non-autoregressive text generation, but their deployment is constrained by large model sizes and heavy computational costs. Post-training quantization (PTQ), a widely used method for compressing and accelerating Large Language Models (LLMs), suffers from severe accuracy degradation and reduced generalization performance when directly applied to DLLMs (e.g., AWQ suffers a 16% accuracy drop on LLADA under W4A4). This paper explores how the unique mechanisms of Dynamic Language Models (DLLMs) conflict with quantization, identifying three core issues: 1) During the iterative generation process of DLLMs, dynamic masking ratios are inherently involved, leading to notable differences in token distributions across decoding steps. Unfortunately, these distinct distributions are not sufficiently captured by current PTQ calibration approaches; 2) Quantization errors propogate and accumalte progressively during iterations in DLLMs, leading to a gradual decline in the performance of quantized models as decoding steps advance; 3) The stability of unmasked tokens, combined with the probabilistic nature of masked tokens, gives rise to an overall feature distribution that is uncoordinated and unsuitable for PTQ. To address these issues, we propose DLLMQuant, a PTQ framework tailored for DLLMs, which incorporates three novel techniques: 1) Temporal-Mask Adaptive Sampling (TMAS), a calibration method that accounts for both time and mask factors, with the capacity to capture distributions across timesteps. 2) Interaction-Aware Activation Quantization (IA-AQ), which utilizes bidirectional attention scores to identify important tokens, and prioritizes these tokens when minimizing quantization error. 3) Certainty-Guided Quantization (CGQ) incorporates mask status and token scores as core weighting criteria for error compensation, enabling PTQ to better align with the unique weight distribution of DLLMs. Experiments show that DLLMQuant achieves significant performance gains (e.g., over 10-point accuracy improvement on GSM8K for LLADA under 4-bit quantization) while enhancing efficiency.

## 1 Introduction

Diffusion-based large language models (DLLMs) have recently attracted growing attention due to their unique advantages and potential applications. Drawing inspiration from diffusion Rombach et al. (2022) processes, they leverage forward masking and reverse recovery to predict masked tokens. By reframing text generation as a denoising task, DLLMs enable parallel decoding while enhancing control over output structure. Notably, they demonstrate strong scalability and even outperform autoregressive-based large language models (LLMs) Kasneci et al. (2023); Bai et al. (2023); Touvron et al. (2023) in specific scenarios—such as addressing the reversal curse Berglund et al. (2023)—highlighting the potential of diffusion models in handling complex language tasks.

However, DLLMs Nie et al. (2025); Ye et al. (2025) still face critical challenges in practical deployment, centered on an intractable efficiency-quality trade-off exacerbated by computational burdens and large model scales. Specifically, simultaneous decoding of multiple tokens tends to degrade generation quality, yet decoding fewer tokens at once leads to a multiplicative increase in the average computational cost per-tokens or even hundreds of times that of autoregressive-based LLMs with

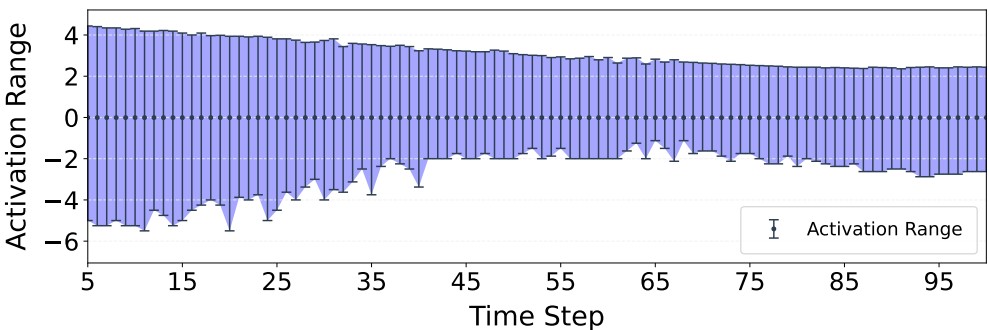

Figure 1: Activation range of outputs from the the first block in LLADA-8B across different time steps, showing significant variations.

comparable scale Bai et al. (2023); Touvron et al. (2023); Dubey et al. (2024). This dilemma arises from two key factors: First, the diffusion mechanism inherent in DLLMs, which initializes an entire response sequence upfront and performs iterative generation with bidirectional attention, resulting in enormous computational overhead. Second, their large parameter sizes, which are comparable to those of autoregressive-based LLMs. To enable sufficient interaction between tokens, DLLMs are designed with feed-forward network (FFN) layers that are even larger than those in autoregressive-based LLMs. Thus, compressing DLLMs and reducing their computational footprint becomes critical for deployment on resource-constrained devices.

Post-Training Quantization (PTQ), which quantizes weights and activations into low-precision formats, effectively reduces memory usage and computational overhead, achieving notable success in LLMs Hu et al. (2025); Xu et al. (2025a;b); Frantar et al. (2022); Xiao et al. (2023). However, directly applying these PTQ approaches to DLLMs leads to substantial performance degradation, particularly in generalization capabilities. For instance, applying AWQ Lin et al. (2024) to LLADA-8B Nie et al. (2025) model leads to more than 16% accuracy decline.

We perform a comprehensive analysis and identify three critical issues. Firstly, DLLMs decode a fixed-length sequence that is initialized entirely with mask tokens through multiple iterations. This iterative process leads to divergent input distributions across time steps. For example, as shown in Fig. 1, feature distributions at early steps differ markedly from those at later ones. This temporal distribution shift poses a significant challenge for PTQ due to the difficulty of capturing distributions across all time steps. Secondly, the iterative generation mechanism introduces another barrier: the output at each time step serves as input for the next prediction, causing quantization errors to propagate and accumulate over iterations. As a result, the performance of quantized models undergoes a progressive decline as iterations proceed. Thirdly, DLLMs employ unique masking and remasking strategies: tokens already decoded remain fixed across iterations, while masked tokens are selectively decoded based on model confidence scores. The evolving process introduces significant disparities in feature distributions across both the token and channel dimensions within certain layers, which undermines the effectiveness of existing PTQ methods relying on GPTQ Lin et al. (2024); Frantar et al. (2022); Ashkboos et al. (2024). This is because GPTQ uniformly treats all tokens when computing the Hessian matrix—used as a weighting factor for error compensation during weight quantization. However, this uniform assumption fails to account for the intrinsic variability in token importance, leading to substantial performance degradation when GPTQ is applied to DLLMs.

To this end, we propose DLLMQuant, an first PTQ framework tailored for DLLMs. DLLMQuant incorporates three novel techniques: 1) Temporal-Mask Adaptive Sampling (TMAS), which is a calibration sampling scheme tailored for the iterative generation process of DLLMs. It captures temporal variations and masking ratio changes during decoding. By strategically selecting calibration data, it restores most of the performance of INT4 quantized models after calibration, emerging as an effective sampling strategy for correcting quantization errors. 2) Interaction-Aware Activation Quantization (IA-AQ), which mitigates the accumulation of errors in iterative steps. Our analysis identifies that quantization of the matrix multiplication following softmax operation in the attention mechanism is a primary source of error propagation. IA-AQ resolves this by computing quantization parameters for the attention module's value matrix via interaction-aware metrics, sharply reducing errors at this critical point. 3) Certainty-Guided Quantization (CGQ), which is a weight quantization

strategy that leverages DLLMs' unique masking and re-masking mechanisms to alleviate the adverse effects of weight quantization. By integrating these three methods, DLLMQuant bridges existing quantization techniques with DLLM architectures, reconciling the performance of quantized systems with the unique requirements of DLLMs. Our contributions are summarized as follows:

- We identify three critical factors that affect the quantization performance of DLLMs: issues in calibration selection, temporal accumulation of quantization errors, and distinct feature distributions induced by unique decoding and re-masking mechanisms.

- We propose TMAS, a calibration scheme adapted to iterative generation in DLLMs; CGQ and IA-AQ, which leverage interaction-aware metrics and certainty guidance to facilitate activation and weight quantization tailored to DLLMs.

- We present DLLMQuant, which seamlessly integrates TMAS, IA-AQ, and CGQ with existing PTQ methods, significantly boosting DLLM quantization performance. As one of the first studies in this domain, we will release the code to facilitate further exploration and advance research in this field.

## 2 RELATED WORK

### 2.1 DIFFUSION-BASED LARGE LANGUAGE MODELS

To address issues such as slow generation speed and reversal curse in autoregressive LLMs, LLaDA Nie et al. (2025) first proposed DLLM. Inspired by diffusion models Croitoru et al. (2023), LLaDA characterizes distributions via two processes: a forward data masking process and a reverse process parameterized by a vanilla Transformer Fedus et al. (2022) to predict masked tokens. The core of LLaDA is a *mask predictor*, a parametric model $p_\theta(\cdot \mid x_t)$ that takes $x_t$ as input and predicts all masked tokens (denoted M) simultaneously. Cross-entropy loss is applied to the masked tokens:

$$\mathcal{L}(\theta) \triangleq -\mathbb{E}_{t,x_0,x_t} \left[ \frac{1}{t} \sum_{i=1}^{L} \mathbf{1}[x_t^i = \mathbf{M}] \log p_\theta(x_0^i | x_t) \right] \tag{1}$$

where $x_0$ is sampled from the training data, $t$ is sampled uniformly from $[0, 1]$, and $x_t$ is sampled from the forward process. The indicator function $\mathbf{1}[\cdot]$ ensures that the loss is computed only for masked tokens. This enables DLLMs to decode multiple tokens simultaneously while maintaining excellent context-aware capabilities. DiffuLLaMA Gong et al. (2024) introduces an ingenious "transformation" approach: it converts pretrained autoregressive models (e.g., LLaMA Touvron et al. (2023)) into DLLMs via adaptive training, significantly reducing the cost compared to training from scratch. LLaDA-1.5 Zhu et al. (2025) successfully applies RLHF-like preference alignment techniques to DLLMs, solving the core problem of large variance in diffusion models' ELBO estimation and significantly improving the model's alignment ability. Multimodal models based on DLLM—such as LaViDa Li et al. (2025) and LLaDA-V You et al. (2025)—have achieved state-of-the-art performance in multimodal understanding tasks, demonstrating the great potential of the end-to-end diffusion paradigm in the multimodal domain.

### 2.2 QUANTIZATION

Quantization involves mapping floating-point numbers to discrete intervals using integer values. When it comes to weight quantization, our focus lies on per-channel symmetric uniform quantization, which is a scheme that has been widely adopted. The quantization process is defined as follows:

$$\mathcal{Q}(\boldsymbol{W}) = \text{clamp}\left(\left\lfloor \frac{\boldsymbol{W}}{s} \right\rceil, q_{\min}, q_{\max}\right) \tag{2}$$

Here, $\boldsymbol{W} \in \mathbb{R}^{oc \times ic}$ denotes the weight matrix, $s \in \mathbb{R}^{oc}$ represents the channel-wise quantization step size, and $q_{\min}, q_{\max}$ specify quantization bounds.

For the quantization of activations, we adopt the widely-used per-tensor asymmetric uniform quantization. The quantization process is expressed as follows:

$$\mathcal{Q}(\boldsymbol{X}) = \text{clamp}\left(\left\lfloor \frac{\boldsymbol{X} - \boldsymbol{z}}{s} \right\rceil, q_{\min}, q_{\max}\right) \tag{3}$$

Here, $\boldsymbol{X} \in \mathbb{R}^{b \times ic}$ denotes the activation matrix, $z$ represents the asymmetric quantization zero point, which is computed as $X_{\min}/s$. For a linear layer, the loss introduced by quantizing both $\boldsymbol{W}$ and $\boldsymbol{X}$ can be formulated as:

$$\mathcal{L}(\boldsymbol{W_q}, \boldsymbol{X_q}) = \|\boldsymbol{W}\boldsymbol{X} - \text{Deq}(\boldsymbol{W_q})\text{Deq}(\boldsymbol{X_q})\|_F^2 \tag{4}$$

Here, $\text{Deq}$ is the de-quantization process, $\boldsymbol{X_q}$ and $\boldsymbol{W_q}$ represent the quantized versions of $\boldsymbol{W}$ and $\boldsymbol{X}$. Notable methods like AWQ Lin et al. (2024) leverage such loss functions to guide selection of smoothing coefficients and weight pruning. GPTQ Frantar et al. (2022) builds on OBQ LeCun et al. (1989), which uses the Hessian matrix to compensate for quantization error. Combined with Eq. 4, the Hessian can be computed as:

$$\boldsymbol{H} = \boldsymbol{X}\boldsymbol{X}^\top \tag{5}$$

## 2.3 POST-TRAINING QUANTIZATION FOR LLMS

Most large language models (LLMs) are constructed on the Transformer Fedus et al. (2022) framework, which is inherently characterized by high memory usage and substantial computational demands. Post-training quantization (PTQ) has established itself as a widely employed strategy for compressing LLMs, as it can effectively cut down memory and computational consumption while maintaining the model's accuracy. Among the various PTQ techniques, GPTQ Frantar et al. (2022) and AWQ Lin et al. (2024) stand out and have undergone extensive research. GPTQ makes use of Hessian-based error compensation to reduce quantization errors, allowing for high compression ratios. AWQ, on the other hand, takes into account how activation distributions influence weight quantization, thereby improving the performance of the quantization process. Beyond these foundational approaches, several advanced techniques have been developed to enhance PTQ further. QuaRot Ashkboos et al. (2024) utilizes Hadamard transformations to get rid of outliers without changing the output, which in turn improves the effectiveness of GPTQ. GPTVQ Van Baalen et al. (2024) delves into non-uniform quantization schemes from a vector viewpoint, providing better adaptability to weight distributions.

However, these methods fail to account for the unique challenges inherent in DLLM architectures, resulting in significant accuracy degradation. Our proposed DLLMQuant, grounded in the interplay between quantization and and the core mechanisms of DLLMs, is orthogonal to existing PTQ approaches. This characteristic enables its seamless integration with prior methods, thereby facilitating the effective quantization of DLLMs.

## 3 METHOD

In this paper, we propose DLLMQuant, a framework designed for efficient quantization of DLLMs. It specifically addresses three core issues: quantization errors accumulate across iterations, distinct token distributions across decoding steps and significant disparities in feature distributions across both token and channel dimensions. DLLMQuant tackles these issues from three aspects: optimizing calibration via Temporal-Mask Adaptive Sampling (TMAS), improving weight quantization with Certainty-Guided Quantization (CGQ), and enhancing activation quantization through Interaction-Aware Activation Quantization (IA-AQ). TMAS generates calibrations with proportionally selected data across time steps and masking ratios, ensuring the quantized model performs well throughout iterative generation. CGQ refines weight quantization compensation by incorporating token mask positions along with their final confidence scores. IA-AQ mitigates quantization error accumulation by leveraging bidirectional attention patterns during activation quantization.

All of the aforementioned solutions are plug-and-play, allowing seamless integration with other quantization techniques to enhance the quantization performance of DLLMs. These solutions are detailed in subsequent sections.

### 3.1 TEMPORAL-MASK ADAPTIVE SAMPLING

Current PTQ methods typically rely on calibration constructed by collecting activation information through random or uniform sampling. While these sampling methods can preserve reasonable generalization capabilities for standard LLMs, their direct application to DLLMs often leads to significant performance degradation. This performance degradation stems from the failure of existing methods to account for two key traits inherent to DLLMs: iterative decoding processes and dynamic masking ratios. These two factors collectively lead to variations in output distributions across different timesteps.

Given that the DLLMs use the same mask prediction network to process inputs at all time steps, determining an effective calibration sampling policy becomes a significant challenge. We begin by analyzing the output distributions of the model's first block across different time steps. Specifically, we conduct an experiment on the LLADA-8B model with 100 denoising steps and 4 blocks, plotting the activation ranges of 1,000 random samples across all time steps on the PIQA Bisk et al. (2020b) dataset. It should be explained that DLLMs divide the total time steps into a number of blocks (such as 4 in this case) and then decode each block sequentially. As shown in Fig. 1, feature distributions gradually change, with neighboring time steps being similar and distant ones being distinctive.

Considering both the high similarity of output distributions across consecutive time steps and the block-based inference decoding mechanism of DLLMs, we propose a time- and mask-aware calibration method. Specifically, as detailed in Alg. 1, we sample inputs at specific intervals and proportions, ensuring they cover diverse masking ratios and span different time steps. This approach has the capability to represent distributions across all time steps. The sampled calibration can restore most of the performance of INT4 quantized models after calibration, rendering it an effective sampling scheme for collecting calibration data in quantization error correction. A detailed discussion of the relevant configurations can be found in Appendix A.3.

---

**Algorithm 1** Temporal-Mask Adaptive Sampling (TMAS)

---

**Require:** Inputs $\mathcal{X}$, Block count $B$, Time steps $T$
**Ensure:** Calibration dataset $\mathcal{D}_c$
1: $s \leftarrow \lfloor T/B \rfloor$ {Steps per block}, $\quad n \leftarrow \lfloor 512/B \rfloor$ { Samples per block}
2: $\mathbf{p} \leftarrow n \cdot [0.3, 0.2, 0.2, 0.3]$ {Target proportion of per mask ratio interval}
3: $\mathbf{C} \leftarrow \text{zeros}(B, 4)$ {Sampling counter matrix}, $\quad \mathcal{D}_c \leftarrow \emptyset$ {Initialize calibration data}
4: **Function** ClassifyMaskRatio($r$):
5: $\quad$ **return** $[0, 1, 2, 3][(r \geq 0.2) + (r \geq 0.5) + (r \geq 0.8)]$
6: **for** $x \in \mathcal{X}$ **do**
7: $\quad$ **for** $t = T - 1$ **to** $0$ **do**
8: $\quad\quad y_t \leftarrow \text{Model}(x)$
9: $\quad\quad r_t \leftarrow |y_t|_{\text{unmasked}} / |y_t|_{\text{total}}$
10: $\quad\quad m \leftarrow \text{ClassifyMaskRatio}(r_t), \quad block \leftarrow \lfloor t/s \rfloor$
11: $\quad\quad$ **if** $\mathbf{C}[block, m] < \mathbf{p}[m]$ **then**
12: $\quad\quad\quad \mathcal{D}_c \leftarrow \mathcal{D}_c \cup \{x\}, \quad \mathbf{C}[block, m] \leftarrow \mathbf{C}[b, m] + 1$
13: $\quad\quad$ **end if**
14: $\quad$ **end for**
15: **end for**

---

### 3.2 INTERACTION-AWARE ACTIVATION QUANTIZATION

Previous research has identified that quantization errors tend to accumulate across layers Dao et al. (2022); Hu et al. (2025), making deeper neural networks more difficult to quantize. In DLLMs, at any time step t, the input to the mask prediction model (denoted as $x_t$) is derived from $x_{t+1}$, which is the model's output at the previous time step $t + 1$. Quantization errors, which inherently accumulate across layers, are further compounded by the number of denoising steps in this iterative process. This leads to a geometric growth of total error as the model progresses through later denoising steps. As shown in Eq. 6, the quantization error $L(X_{t+1})$ at time step $t + 1$ propagates through the model's iterations to time step $t$, causing a further increase in the quantization error $L(x_t)$ at time step $t$. Here, $\mathcal{Q}_{\text{model}}$ denotes the quantized model, and Deq represents the dequantization operation.

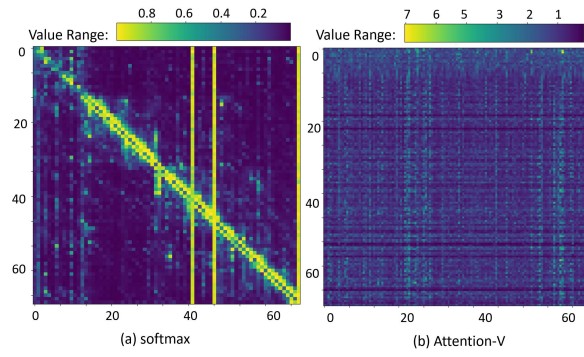

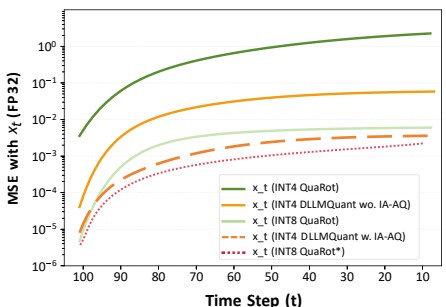

Figure 2: Output distributions within LLADA-8B attention in Block.1.layer.0. The softmax output (a) is notably sparse, whereas the value matrix (b) exhibits significant distribution discrepancies across channels and tokens.

Figure 3: Cumulative quantization error of LLADA-8B over time steps under different methods (QuaRot* denotes that matmul after softmax operation in attention remains unquantized.)

$$L(x_t) = x_t - \mathrm{Deq}(\mathcal{Q}(x_t + L(x_{t+1})))$$
$$= \mathcal{Q}_{\mathrm{Model}}(x_{t+1}) - \mathcal{Q}_{\mathrm{Model}}\big(\mathrm{Deq}(\mathcal{Q}(x_{t+1}))\big) \tag{6}$$

We conduct experiments using the LLADA-8B model on the PIQA dataset, comparing the mean squared error (MSE) differences between the full-precision model and models quantized to INT8 and INT4 using different methods at each time step. As illustrated in Fig. 3, when the model is quantized to 4-bit using QuaRot, quantization errors increase significantly during iteration. This makes it difficult to preserve the performance of the model.

Fortunately, our experiments reveal a key factor in error accumulation under low-bit quantization. Specifically, this factor refers to the quantization error introduced by the matrix multiplication (matmul) between the output of the softmax operation and value matrix in attention. In the case of INT8 QuaRot quantization, we conduct comparative experiments where this specific component was either quantized or left unquantized. As shown in Fig. 3, the accumulated error when this component remains unquantized is significantly lower than when it is quantized.

To explore this further, we visualize the output features of this component, as shown in Fig. 2. We observe that the distribution of the value matrix V exhibits substantial variability across both token and channel dimensions, which poses issues for quantization. Additionally, the softmax output exhibits pronounced sparsity: larger values are concentrated near the diagonal and only within a small subset of tokens, while the rest are negligible. This phenomenon is closely tied to the unique interaction mechanisms of DLLMs, including their carefully designed bidirectional attention and large key-value (KV) heads. These features enable sufficient token interaction, endowing DLLMs with reverse reasoning and contextual awareness.

$$\mathcal{L}(s) = \left\| \left( \left\lfloor \frac{V - z}{s} \right\rceil - V \right) \cdot \mathrm{Deq}(O_{\mathrm{softmax}}) \right\|_F^2 \tag{7}$$

To mitigate the cumulative errors arising from quantization, we propose Interaction-Aware Activation Quantization (IA-AQ). Specifically, as described in Eq. 7, when calculating the quantization parameters for value matrix $V$ prior to matrix multiplication, we redesign the quantization error metric by treating the softmax output as a weighting term. In Eq. 7, $z$ denotes the zero-point, $s$ represents the scaling factor and $O_{\mathrm{softmax}}$ represents the output of the quantized softmax function. To determine the optimal scaling factor $s$, we begin with the standard quantization scaling $\hat{s}$ and test $\alpha$ values (stepping by 0.02 from 1.0 to 0.8) to minimize $L(\alpha \odot \hat{s})$:

$$\hat{s} = (V_{\max} - V_{\min})/(Q_{\max} - Q_{\min}) \tag{8}$$

$$s = \alpha \odot \hat{s} = \underset{\alpha \in \{1.0, 0.8\}}{\arg\min} \ L(\alpha \odot \hat{s}) \tag{9}$$

This approach dynamically allocates quantization resources to suppress interference from features tied to tokens with weak interactions, while ensuring accurate quantization of features associated with tokens exhibiting both high interaction frequency and critical importance.

### 3.3 CERTAINTY-GUIDED QUANTIZATION

As previously described, DLLMs perform iterative decoding with a fixed input-output length. Tokens that have been unmasked remain unchanged in subsequent iterations, while the masked portion is decoded based on the model's final output scores. Therefore, treating masked and unmasked tokens equally during quantization is inappropriate. Specifically, as illustrated in Fig. 4, errors in unmasked or low-score regions do not propagate through iterations and thus do not affect subsequent decoding steps.

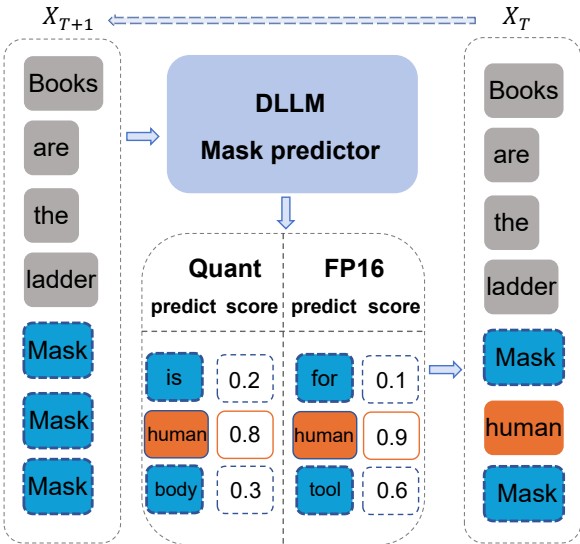

We analyze the statistical distribution of output scores and find that only a small subset of tokens have relatively high scores, while the majority exhibit low scores. Notably, tokens with high scores are precisely those decoded in the current iteration, and their variations directly influence the input for the next iteration. In contrast, both low-score tokens and already decoded tokens do not affect subsequent iterations—the positions corresponding to low-score tokens remain masked in the following step.

Figure 4: DLLMs iterative inference with masking and remasking strategies. The quantization errors of unmasked tokens (gray boxes) and masked tokens with low confidence scores (blue boxes) do not affect the input of the next iteration.

Hessian-based PTQ methods typically quantize weights column-wise and adjust subsequent unquantized columns using statistically computed Hessian matrices to compensate for already quantized ones. However, conventional Hessian statistical computation fails to account for the aforementioned characteristics of DLLMs, leading to suboptimal performance. Based on this insight, we propose the Certainty-Guided Quantization (CGQ) method for optimizing weight quantization. Specifically, during quantization, we place greater emphasis on the unmasked regions with higher scores. In implementation, CGQ leverages weighted Hessian matrices to guide compensation during weight quantization. As shown in Eq. 10, when computing Hessian matrix, CGQ integrates coefficients derived from mask regions and token final scores, thereby guiding weight updates to minimize quantization errors.

$$H = (X \odot (\mathbf{1}[X_t = M] + \sqrt{sc_t})) \times (X \odot (\mathbf{1}[X_t = M] + \sqrt{sc_t}))^\top \tag{10}$$

Here, $\mathbf{1}[X_t = M]$ is a custom weighted indicator function. Specifically, for masked regions (i.e., regions where $X_t = M$), it assigns a weight of 1; for unmasked regions (i.e., regions where $X_t \neq M$), it assigns a weight of 0.7. Here, $sc_t$ denotes the final confidence score assigned to each token in model output. Notably, unmasked regions, though no longer updated, remain non-negligible due to their interaction with masked regions that provide contextual information. Thus, we assign weights of 0.7 to unmasked segments. We acknowledge that more refined parameter tuning may yield better results; however, the current configuration already enables CGQ to effectively account for token masking states and final scores, thereby optimizing weight quantization and improving the performance of quantized models.

## 4 EXPERIMENTS

In this section, we first describe the experimental setup, including the models, datasets, and baselines. We then present the results of comparative experiments across diverse datasets to validate the

| Model | Method | Truth. | Arc. | Hella. | Wino. | PIQA | MMLU | C-EVAL | Hum. | GSM8K | Avg. |
|---|---|---|---|---|---|---|---|---|---|---|---|
| LLADA | FP | 47.49 | 44.03 | 54.06 | 74.9 | 74.65 | 65.85 | 69.54 | 32.92 | 67.48 | 59.87 |
| | RTN | 40.45 | 41.83 | 45.40 | 64.72 | 67.95 | 49.26 | 57.95 | 14.02 | 16.56 | 44.23 |
| | AWQ | 40.87 | 42.92 | 46.14 | 66.88 | 69.43 | 51.22 | 58.43 | 20.10 | 36.88 | 48.09 |
| | DLLMQuant$^+$ | 41.53 | 43.44 | 46.51 | 67.87 | 70.12 | 51.72 | 59.38 | 22.13 | 40.66 | 49.26 |
| | QuaRot | 42.53 | **44.20** | 49.76 | 69.85 | 70.75 | 55.96 | 56.32 | 25.33 | 44.57 | 51.03 |
| | DLLMQuant$^{++}$ | **43.53** | 44.18 | **51.00** | **71.85** | **73.94** | **57.77** | **61.22** | **28.92** | **56.25** | **54.29** |
| LLADA-1.5 | FP | 47.2 | 88.5 | 74.7 | 74.8 | 74.86 | 66.0 | 70.05 | 49.4 | 83.3 | 69.86 |
| | RTN | 39.51 | 81.77 | 56.82 | 65.27 | 66.91 | 48.88 | 58.96 | 23.44 | 36.56 | 53.12 |
| | AWQ | 40.96 | 82.22 | 66.84 | 67.93 | 69.32 | 51.12 | 60.03 | 30.07 | 57.95 | 58.49 |
| | DLLMQuant$^+$ | 42.14 | 83.38 | **70.09** | 68.69 | 70.22 | 51.63 | 61.28 | 32.44 | 59.55 | 59.94 |
| | QuaRot | 43.21 | **84.23** | 65.34 | 69.55 | 70.17 | 56.23 | 57.66 | 37.33 | 65.86 | 61.06 |
| | DLLMQuant$^{++}$ | **43.87** | 84.18 | 69.20 | **71.74** | **73.58** | **57.27** | **60.04** | **44.58** | **74.33** | **64.31** |
| DREAM | FP | 49.76 | 59.80 | 73.30 | 74.50 | 75.66 | 69.5 | 64.89 | 57.9 | 77.2 | 66.94 |
| | RTN | 41.25 | 54.34 | 58.32 | 64.52 | 66.26 | 51.51 | 49.89 | 28.90 | 30.82 | 49.53 |
| | AWQ | 43.66 | 57.82 | 65.57 | 67.13 | 68.96 | 55.5 | 53.27 | 33.14 | 48.98 | 54.89 |
| | DLLMQuant$^+$ | 44.14 | 58.38 | 66.93 | 68.66 | 69.53 | 57.63 | 54.21 | 35.12 | 51.14 | 56.19 |
| | QuaRot | 47.58 | 58.18 | 67.13 | 70.05 | 70.36 | 69.5 | 53.19 | 34.48 | 59.20 | 58.85 |
| | DLLMQuant$^{++}$ | **47.86** | **59.43** | **70.14** | **71.54** | **72.09** | **69.5** | **55.89** | **44.50** | **66.17** | **61.90** |

Table 1: Results of RTN, AWQ, QuaRot, and ours DLLMQuant with 4-bit quantization for both weight and activation among 9 tasks on LLADA-8B, LLADA-1.5-8B, DREAM-7B. DLLMQuant$^+$ denotes DLLMQuant based on AWQ, and DLLMQuant$^{++}$ denotes DLLMQuant based on QuaRot.

robustness of DLLMQuant. In addition, we conduct ablation studies and analyze the speed of both float16 and quantized models.

## 4.1 SETUP

We adopt symmetric uniform quantization for weights and asymmetric uniform quantization for activations in DLLMs. Specifically, weight quantization is performed with per-channel granularity, while activation quantization uses per-token granularity. All experiments are conducted on NVIDIA A6000 GPUs, unless otherwise specified. As DLLMQuant is an efficient post-training quantization (PTQ) framework, it eliminates the need for any fine-tuning.

**Models and Datasets.** We conducted experiments on the LLADA-8B Nie et al. (2025), LLADA-1.5-8B Zhu et al. (2025), and DREAM-7B Ye et al. (2025) models. Following the testing methods in the LLADA paper, we evaluate the accuracy metric on TruthfulQA-MC2 Lin et al. (2021), Arc-Challenge Clark et al. (2018), HellaSwag Zellers et al. (2019), WinoGrande Sakaguchi et al. (2021), PIQA Bisk et al. (2020a), MMLU Hendrycks et al. (2021), and C-EVAL Huang et al. (2023). Furthermore, we also evaluate DLLMQuant using HumanEval Chen et al. (2021) and GSM8k Cobbe et al. (2021). HumanEval evaluates code generation capabilities, while GSM8k assesses multistep mathematical reasoning skills.

**Baseline** Our primary baselines consist of vanilla RTN and the PTQ methods for LLMs: AWQ Lin et al. (2024) and QuaRot Ashkboos et al. (2024). For calibration, 128 segments from the WinoGrande dataset are selected. Floating-point results are provided as references. For QuaRot, following the implementation method in the official repository, we adopted the GPTQ method—a weight compensation approach based on Hessian matrices—to compensate for the quantized weights. We use the accuracy testing methods provided in the official LLADA repository. In experiments involving AWQ, we adapt its official repository to support the three DLLMs.

## 4.2 RESULTS

**Comparison results.** We comprehensively compare quantization performance across various DLLMs and tasks. As shown in Tab. 1, results from nine tasks demonstrate that our DLLMQuant outperforms other methods on DLLMs, achieving the highest accuracy on nearly all tasks. Across the three DLLMs, it outperforms the original methods by an average of 2% based on the nine-task mean score. On some tasks such as TruthfulQA-MC2 and Arc-Challeng, the results based on QuaRot are not far behind DLLMQuant. However, on the HumanEval and GSM8k tasks, other methods like

QuaRot degrade the model's reasoning ability after quantization. In contrast, DLLMQuant effectively preserves the reasoning ability in generation tasks, achieving results comparable to full-precision models. This is particularly important as reasoning in complex tasks such as HumanEval is crucial for real-world applications, further highlighting the practical relevance of DLLMQuant's performance.

**Ablation results.** DLLMQuant improves the quantization performance of DLLMs through three primary methods: TMAS, CGQ, and IA-AQ. To evaluate these methods, we conduct decomposition experiments. As can be seen in Tab.( 7), the addition of each individual method yields better metrics than when that method is not included.

In addition, as presented in Tab. 2, we performed ablation experiments on the two key components of the CGQ method — the mask state and score (i.e., whether to incorporate the corresponding parts in Eq. 10). From the results, it is evident that considering either of these two factors alone fails to achieve performance as good as considering both factors jointly. When both the mask and score are integrated, the model attains the highest performance on both the GSM8K and HumanEval benchmarks, which validates the complementary effects of these two components in the CGQ method.

| Method | GSM8K | Hum. |
|---|---|---|
| QuaRot | 44.57 | 25.33 |
| QuaRot+ CGQ with mask | 45.07 | 25.76 |
| QuaRot+ CGQ with score | 45.22 | 26.06 |
| QuaRot+ CGQ with score & mask | **45.58** | **26.85** |

Table 2: Ablation Study of Component Combinations in CGQ for 4-bit quantization on LLAMA-8B.

**Memory and Speedup.** The core motivation of DLLMQuant lies in compressing diffusion-based large language models to a lower bitwidth, which aims to reduce both inference latency and GPU memory consumption while maximizing accuracy retention, thus ensuring practical applicability. As presented in Tab. 3, DLLMQuant achieves an average inference speedup of over 1.6× and memory savings exceeding 3.2×, marking substantial improvements in inference efficiency. These advancements facilitate the deployment of DLLMs on consumer-grade devices such as the Nvidia 4090 GPU.

| MODEL | Speed (Tokens/s) | | | Memory (GB) | | |
|---|---|---|---|---|---|---|
| | FP | Quant | Speed Up | FP | Quant | Mem. Sav. |
| LLADA | 34.59 | 50.14 | **1.71** | 15.89 | 4.91 | **3.24** |
| LLADA-1.5 | 35.55 | 60.43 | **1.70** | 15.88 | 4.90 | **3.24** |
| DREAM | 23.27 | 35.84 | **1.54** | 13.95 | 4.44 | **3.14** |

Table 3: Speedup and memory saving of three DLLMs, compared between our 4-bit implementation and FP16.

## 5 CONCLUSION

In this paper, we address the critical challenge of quantizing diffusion-based large language models (DLLMs). DLLMs feature unique mechanisms, including iterative generation, dynamic masking, and bidirectional attention. Conventional PTQ methods, while effective for standard LLMs, perform poorly when directly applied to DLLMs. We identify three core issues behind this failure: existing calibration methods fail to capture token distributions that vary with time steps and masking ratios; quantization errors accumulate and amplify across iterations; and conventional quantization strategies mismatch DLLM feature distributions, where fixed unmasked tokens coexist with probabilistic masked tokens. To address these issues, we propose DLLMQuant, a novel PTQ framework tailored for DLLMs, integrating three key techniques: Temporal-Mask Adaptive Sampling, which balances the size and representativeness of calibration, ensuring robust quantization throughout iterations. Interaction-Aware Activation Quantization, which mitigates error accumulation by dynamically allocating quantization resources to critical attention modules, particularly targeting matrix multiplication in the attention mechanism. Certainty-Guided Quantization, which boosts weight quantization by giving priority to error compensation for high-confidence masked tokens (tokens that are about to be decoded) and integrating information from token mask states. Experiments on DLLMs demonstrate DLLMQuant outperforms baselines, with top accuracy on most tasks, preserved reasoning capabilities, and 2% average gains. DLLMQuant bridges PTQ methods and DLLM architectures, enabling efficient compression and acceleration without significant accuracy degradation.

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

# A APPENDIX

## A.1 LLM DISCLAIMER

The authors gratefully acknowledge the support of large language model (LLM) tools, which have assisted in refining the paper's text and optimizing its expression.

## A.2 OVERALL RESULTS OF ABLATION RESULTS

Tab. 4 presents the complete results of the ablation study on the proposed TMAS, CGQ, and IA-AQ based on AWQ/GPTQ baselines. It evaluates the average performance (Avg.) of 4-bit weight and activation quantization for the LLADA model across nine tasks.

Tab. 7 presents the performance of 4-bit weight and activation quantization for the LLADA model on GSM8K and HumanEval tasks, where the calibration sets are constructed based on different sampling methods and the quantization is based on RTN. Here, LLMQAT Liu et al. (2023) employs a self-generated calibration approach.

Tab. 6 illustrates the results of various quantization methods, including RTN, GPTQ, SmoothQuant, OSTQuant, and our DLLMQuant, with 4-bit weight and activation quantization on LLADA-8B and LLADA-1.5-8B across 9 tasks. When using 4-bit quantization, our DLLMQuant variants (DLLMQuant*, DLLMQuant**, DLLMQuant*+) demonstrate notable advantages. Compared to other methods like RTN, GPTQ, and SmoothQuant, DLLMQuant variants achieve higher average scores on both LLADA and LLADA-1.5 models. For instance, on LLADA-1.5, DLLMQuant*+ attains an average score of 65.21, which is substantially higher than the averages of many other methods. This shows that DLLMQuant can better preserve model performance under 4-bit quantization, effectively balancing the need for reduced memory and computational costs with maintaining high task-solving capabilities, outperforming existing quantization techniques in terms of accuracy retention.

| Method | GSM8K | HumanEval |
|---|---|---|
| RTN + Random_calib | 16.56 | 14.02 |
| RTN + LLMQAT | 15.43 | 13.34 |
| RTN + Uniform_time | 17.44 | 15.82 |
| RTN + TMAS | 18.12 | 16.56 |

Table 5: The performance of 4-bit weight and activation quantization for the LLADA under different sampling methods.

| Model | Method | Truth. | Arc. | Hel. | Wino. | PIQA | MMLU | C-EVAL | Hum. | GSM8K | Avg. |
|---|---|---|---|---|---|---|---|---|---|---|---|
| LLADA | AWQ | 40.87 | 42.92 | 46.14 | 66.88 | 69.43 | 51.22 | 58.43 | 20.10 | 36.88 | 48.09 |
| | AWQ + TMAS + IA-AQ | 41.12 | 43.22 | 46.29 | 67.41 | 70.12 | 51.48 | 59.25 | 21.07 | 37.92 | 48.55 |
| | AWQ + CGQ + IA-AQ | 41.33 | 43.02 | 46.31 | 67.12 | 70.03 | 51.52 | 59.13 | 21.18 | 38.12 | 48.63 |
| | AWQ + TMAS + CGQ | 41.37 | 43.12 | 46.35 | 67.68 | 70.23 | 51.46 | 59.47 | 22.10 | 39.78 | 49.06 |
| | AWQ + TMAS + CGQ + IA-AQ | 41.53 | 43.44 | 46.51 | 67.87 | 70.12 | 51.72 | 59.38 | 22.13 | 40.66 | 49.26 |
| | QuaRot | 42.53 | 44.20 | 49.76 | 69.85 | 70.75 | 55.96 | 56.32 | 25.33 | 44.57 | 51.03 |
| | QuaRot + TMAS + IA-AQ | 43.04 | 44.09 | 50.11 | 70.93 | 71.77 | 56.34 | 58.13 | 27.92 | 48.23 | 53.36 |
| | QuaRot + CGQ + IA-AQ | 43.18 | 43.92 | 50.05 | 70.85 | 71.85 | 56.56 | 58.33 | 27.87 | 48.63 | 52.36 |
| | QuaRot + TMAS + CGQ | 43.33 | 44.17 | 50.76 | 71.25 | 72.34 | 56.96 | 59.47 | 28.04 | 52.18 | 53.16 |
| | QuaRot + TMAS + CGQ + IA-AQ | 43.53 | 44.18 | 51.00 | 71.85 | 73.94 | 57.77 | 61.22 | 28.92 | 56.25 | 54.29 |

Table 4: The complete results of the ablation study on the proposed TMAS, CGQ, and IA-AQ based on AWQ/GPTQ baselines.

| Model | Method | Truth. | Arc. | Hella. | Wino. | PIQA | MMLU | C-EVAL | Hum. | GSM8K | Avg. |
|---|---|---|---|---|---|---|---|---|---|---|---|
| | FP | 47.49 | 44.03 | 54.06 | 74.9 | 74.65 | 65.85 | 69.54 | 32.92 | 67.48 | 59.87 |
| | RTN | 40.45 | 41.83 | 45.40 | 64.72 | 67.95 | 49.26 | 57.95 | 14.02 | 16.56 | 44.23 |
| | GPTQ | 41.07 | 42.78 | 47.10 | 66.78 | 69.85 | 52.45 | 58.66 | 21.03 | 38.81 | 48.73 |
| LLADA | DLLMQuant* | 41.85 | 43.76 | 47.21 | 67.87 | 71.14 | 53.37 | 60.05 | 23.21 | 42.27 | 50.75 |
| | SmoothQuant | 41.11 | 42.69 | 46.95 | 66.92 | 70.03 | 52.54 | 58.63 | 21.31 | 38.91 | 48.79 |
| | DLLMQuant** | 41.92 | 43.58 | 47.05 | 68.14 | 71.02 | 53.77 | 60.44 | 23.45 | 42.66 | 50.11 |
| | OSTQuant | 45.13 | 44.20 | 50.25 | 71.65 | 72.18 | 58.34 | 59.32 | 29.13 | 60.57 | 54.53 |
| | DLLMQuant*+ | 45.24 | 44.15 | 52.01 | 71.93 | 72.65 | 60.04 | 60.32 | 30.13 | 61.27 | 55.30 |
| | FP | 47.2 | 88.5 | 74.7 | 74.8 | 74.86 | 66.0 | 70.05 | 49.4 | 83.3 | 69.86 |
| | RTN | 39.51 | 81.77 | 56.82 | 65.27 | 66.91 | 48.88 | 58.96 | 23.44 | 36.56 | 53.12 |
| | GPTQ | 41.15 | 81.41 | 67.34 | 68.54 | 69.82 | 51.52 | 60.40 | 31.61 | 58.24 | 58.89 |
| LLADA-1.5 | DLLMQuant* | 42.65 | 83.77 | 70.11 | 70.69 | 70.04 | 51.63 | 61.52 | 33.24 | 60.03 | 60.41 |
| | SmoothQuant | 41.33 | 82.13 | 66.94 | 68.65 | 69.90 | 52.21 | 60.62 | 31.71 | 59.26 | 59.19 |
| | DLLMQuant** | 42.72 | 84.10 | 70.33 | 70.75 | 70.21 | 52.17 | 61.68 | 34.54 | 61.86 | 60.93 |
| | OSTQuant | 45.32 | 85.11 | 68.33 | 70.07 | 71.32 | 55.32 | 65.79 | 43.31 | 73.14 | 64.19 |
| | DLLMQuant*+ | 45.41 | 85.02 | 68.45 | 72.12 | 72.20 | 54.45 | 67.71 | 45.31 | 76.18 | 65.21 |

Table 6: Results of RTN, GPTQ, SmoothQuant, OSTQuant Hu et al. (2025),and ours DLLMQuant with 4-bit weight and activation quantization among 9 tasks on LLADA-8B, LLADA-1.5-8B) . DLLMQuant* denotes DLLMQuant based on GPTQ, DLLMQuant* denotes DLLMQuant based on SmoothQuant and DLLMQuant*+ denotes DLLMQuant based on OSTQuant.

| Proportion | num | GSM8K | HumanEval |
|---|---|---|---|
| [0.25, 0.25, 0.25, 0.25] | 256 | 55.65 | 28.13 |
| [0.3, 0.2, 0.2, 0.3] | 256 | 55.91 | 28.42 |
| [0.2, 0.3, 0.3, 0.2] | 256 | 55.51 | 27.92 |
| [0.2, 0.2, 0.2, 0.4] | 256 | 55.56 | 27.96 |
| [0.4, 0.2, 0.2, 0.2] | 256 | 55.62 | 28.02 |
| [0.25, 0.25, 0.25, 0.25] | 512 | 55.77 | 28.36 |
| [0.2, 0.3, 0.3, 0.2] | 512 | 55.62 | 28.14 |
| [0.3, 0.2, 0.2, 0.3] | 512 | 56.25 | 28.92 |
| [0.2, 0.2, 0.2, 0.4] | 512 | 55.85 | 28.33 |
| [0.4, 0.2, 0.2, 0.2] | 512 | 55.73 | 28.17 |
| [0.25, 0.25, 0.25, 0.25] | 1024 | 55.92 | 28.26 |
| [0.2, 0.3, 0.3, 0.2] | 1024 | 55.56 | 28.22 |
| [0.3, 0.2, 0.2, 0.3] | 1024 | 56.27 | 28.03 |
| [0.2, 0.2, 0.2, 0.4] | 1024 | 55.79 | 28.12 |
| [0.4, 0.2, 0.2, 0.2] | 1024 | 55.46 | 28.13 |

Table 7: The performance of 4-bit weight and activation quantization for the LLADA under different sampling configurations in Quarot-based DLLMQuant.

## A.3 Ablation Experiments for Configuration Settings

As shown in Alg. 1, we set the sampling proportion as [0.3, 0.2, 0.2, 0.3] and the sampling number as 512. Through multiple sets of ablation and control experiments, we found that on the basis of uniform distribution, slightly increasing the weight of the start and end time steps leads to better performance. As for the sampling number, there is a trend that a larger number brings better results. For example, when the number increases from 256 to 512, the performance improves notably, as can be seen from the values in the GSM8K and HumanEval columns in the table (e.g., for the proportion [0.3, 0.2, 0.2, 0.3], GSM8K score rises from 55.91 to 56.25 and HumanEval score goes from 28.42 to 28.92). However, the gain from increasing the number from 512 to 1024 is not particularly obvious. Taking the same proportion [0.3, 0.2, 0.2, 0.3] as an example, the GSM8K score only increases from 56.25 to 56.27 and the HumanEval score changes from 28.92 to 28.03, indicating a marginal improvement. While more granular parameter tuning might potentially achieve better results, our

756
757
758
759
760
761
762
763
764
765
766
767
768
769
770
771
772
773
774
775
776
777

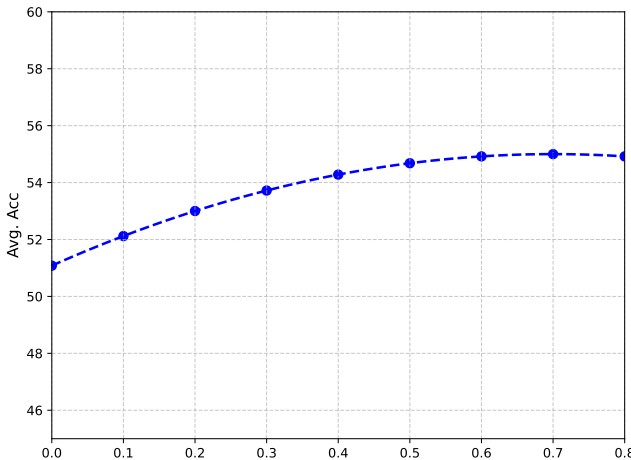

Figure 5: Ablation Study on the Weight of Unmasked Regions for the LLADA in Quarot-based DLLMQuant.

778
779
780
781
782
783
784
785
786
787
788
789
790

chosen configuration already enables TMAs to perceive temporal differences in DLLMs during sampling, thereby reducing quantization errors.

In the CGQ method, Equation 10 assigns a weight of 1 to masked regions (i.e., where $X_t = M$) and a weight of 0.7 to unmasked regions (i.e., where $X_t \neq M$). Here, $sc_t$ represents the final confidence score allocated to each token in the model's output. Notably, despite no longer being updated, unmasked regions retain significance due to their interactions with masked regions, which supply critical contextual information.To validate the appropriateness of the 0.7 weight, we conducted ablation experiments varying this parameter. As illustrated in the Fig. 5, when adjusting the weight value (plotted on the x-axis), the average accuracy (Avg Acc, y-axis) exhibits a distinct upward trend before stabilizing at a peak performance level around the 0.7 value. This clearly indicates that 0.7 represents a well-suited weight for unmasked regions. While more granular parameter tuning might potentially yield marginal improvements, our selected configuration effectively enables the quantization process to perceive and account for mask states, ensuring robust performance.

791
792

A.4 OUTPUT DISTRIBUTION OF SPECIFIC LAYERS IN DLLMS

793
794
795
796

Fig. 6 shows the distribution of softmax output in different blocks of LLADA. It can be observed that, across the entire model, the softmax outputs exhibit a relatively obvious sparsity. Except for the areas near the diagonal and some individual tokens with larger values, the values in other regions are very small.

797
798
799
800

Fig. 7 shows per-channel distribution of the FFN outputs in the attention mechanism of the first block of LLADA. It can be observed that there is an obvious difference in value distribution between the first iteration (i.e., (a) in the figure) and the last iteration (i.e., (c) in the figure).

801
802
803
804
805
806
807
808
809

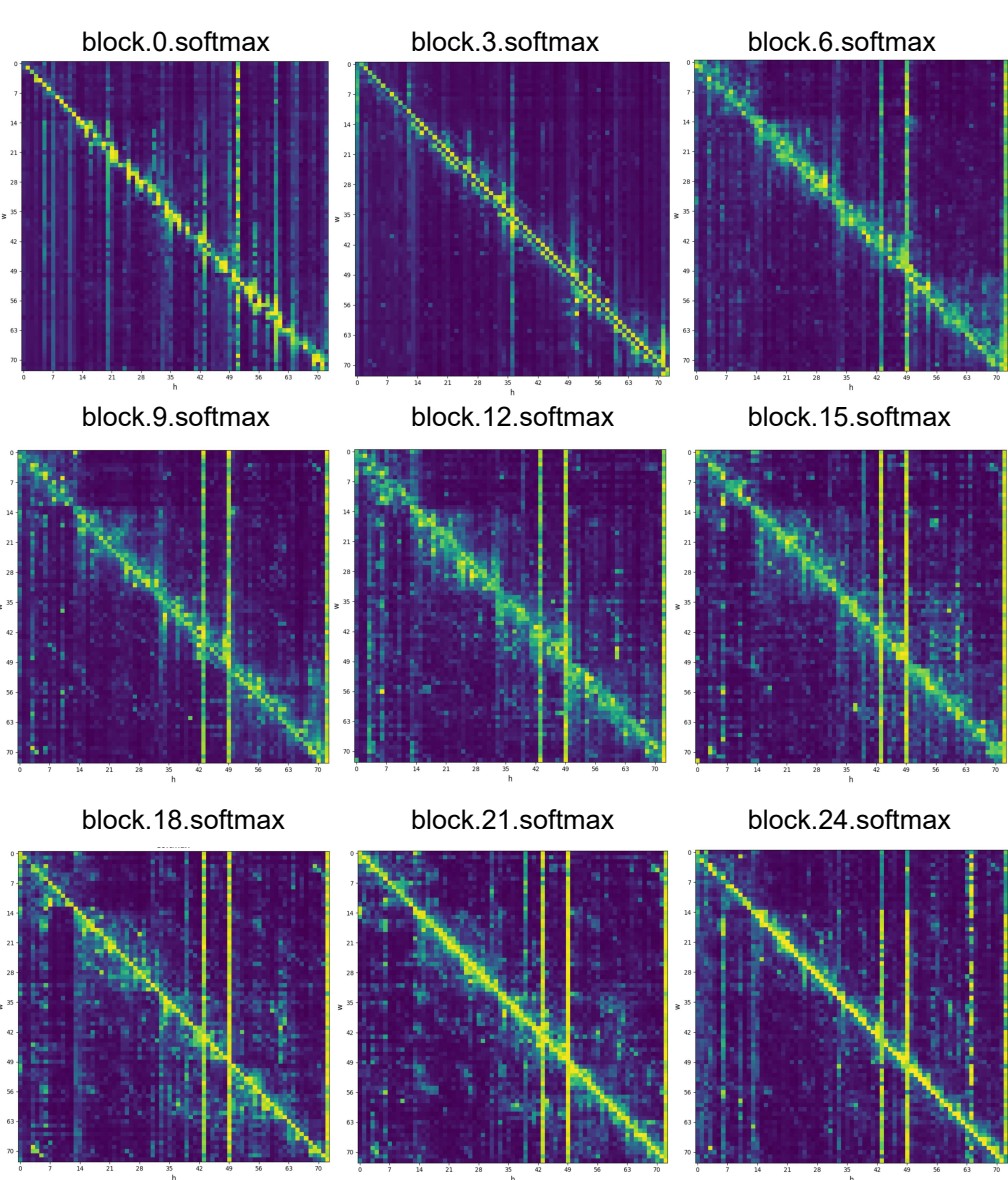

Figure 6: Output distribution of softmax in different blocks of LLADA. The softmax output exhibits pronounced sparsity: larger values are concentrated near the diagonal and only within a small subset of tokens, while the rest are negligible. This phenomenon is closely tied to the unique interaction mechanisms of DLLMs, including their carefully designed bidirectional attention and large key-value (KV) heads.

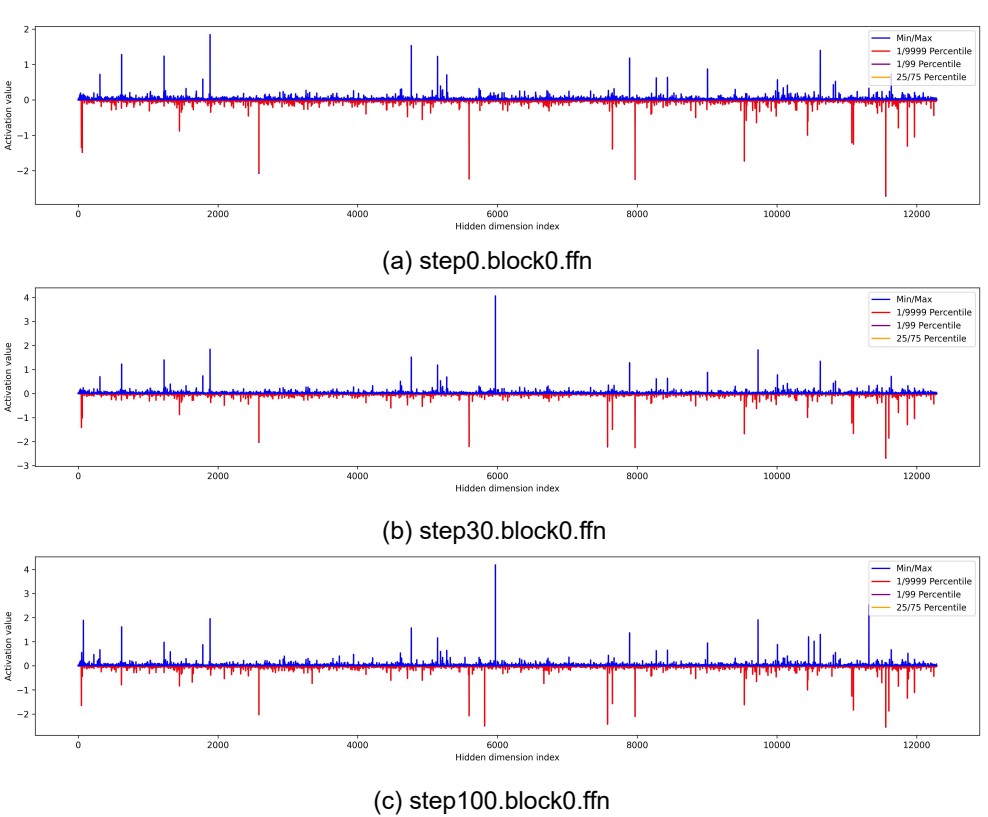

(a) step0.block0.ffn

(b) step30.block0.ffn

(c) step100.block0.ffn

Figure 7: The per-channel output distribution of ffn operation in the attention mechanism across different iteration steps of LLADA. It can be observed that the magnitude of the output values varies significantly across the hidden dimension, with substantial differences among individual channels.

