# OpenReview forum: "DLLMQuant: A Post-Training Quantization Framework Tailored for Diffusion-Based Large Language Models"
_ICLR.cc/2026/Conference — Submitted to ICLR 2026_

### Official Review · Reviewer_8GDD · 2025-10-29

**Soundness:** 1
**Presentation:** 3
**Contribution:** 2
**Rating:** 2
**Confidence:** 4

**Summary:**

To reduce the memory and computational cost of diffusion-based large language models (DLLMs), the authors propose a post-training quantization (PTQ) method.
They observe that directly applying conventional PTQ methods to DLLMs leads to significant performance degradation due to three main challenges.
First, to address the data distribution shift across time steps, the authors introduce a Temporal-Mask Adaptive Sampling strategy that collects calibration data from various time steps and masking ratios, effectively capturing the diverse data distributions present during inference.
Second, to mitigate the accumulation of quantization errors across iterations, the authors identify that these errors mainly arise from quantized matrix multiplications of the value matrix and the softmax outputs. To alleviate this issue, they propose an Interaction-Aware Activation Quantization method that searches for the optimal quantization parameters of the value matrix, reducing cumulative quantization error.
Third, recognizing the different impacts of masked and unmasked tokens to the model output, the authors assign different importance factors to masked and unmasked tokens when profiling the Hessian matrices of weights, leading to more accurate quantization sensitivity estimation.
Experimental results demonstrate that the proposed method achieves an average 3% performance improvement over state-of-the-art baselines under 4-bit weight and activation quantization.

**Strengths:**

1. The paper provides a comprehensive and well-organized background on diffusion-based large language models (LLMs) and quantization techniques, making the work accessible and informative for readers who may not be deeply familiar with these topics.
2. The paper clearly describes the motivation behind each design choice by contrasting autoregressive LLMs with diffusion-based LLMs to justify the necessity of the proposed method.
3. The proposed method demonstrates notable improvements in both algorithmic performance over baseline models and hardware efficiency compared to the original model when deployed on consumer-level GPUs, underscoring its practical applicability and potential for real-world deployment.

**Weaknesses:**

1. The motivation of Interaction-Aware Activation Quantization (IA-AQ) could be further clarified and strengthened.

$~$ (a) In Figure 3, using unquantized matrix multiplication in the attention module helps reduce quantization error. However, the error still appears to accumulate across time steps, suggesting that the proposed approach may not address the issue of quantization error accumulation. It would be helpful if the authors could explain how IA-AQ aims to mitigate this problem theoretically or using experimental results.

$~$ (b) The relationship between searching for scaling factors and enabling sufficient token interaction remains somewhat ambiguous. A more detailed explanation of this relationship would strengthen the rationale of the proposed design.

2. The overall workflow of the proposed method could be presented more clearly.

$~$ (a) In Equation (10), the authors compute the Hessian matrices of the weights, but it is not fully explained how the Hessian matrices is utilized in subsequent steps. Providing a pseudo-code or schematic diagram of the full pipeline would greatly improve readability and understanding.

$~$ (b) In Tables 1 and 2, the proposed CGQ is combined with AWQ[1] and QuaRot[2] to validate its effectiveness. Since AWQ and QuaRot do not require Hessian matrices while CGQ refines their calculation, clarifying the implementation details would avoid confusion.

3. Some parts of the manuscript appear to be incomplete.

$~$ (a) There is a placeholder “TODO” table in line 439, indicating missing ablation results for the different components of DLLMQuant.

$~$ (b) Figure 4 contains meaningless notations.


[1] Lin, Ji, et al. "Awq: Activation-aware weight quantization for on-device llm compression and acceleration." Proceedings of machine learning and systems 6 (2024): 87-100.

[2] Ashkboos, Saleh, et al. "Quarot: Outlier-free 4-bit inference in rotated llms." Advances in Neural Information Processing Systems 37 (2024): 100213-100240.

**Questions:**

See weaknesses.

---

> ### Author Response · Authors · 2025-11-13
> **Rebuttal by Authors**
>
> We sincerely thank you for your valuable time and efforts in reviewing our manuscript. We have addressed each comment and made the necessary revisions to improve the quality and clarity of our manuscript.
>
> **1. Clarification on IA-AQ's Mechanism for Mitigating Quantization Error Accumulation**
>
> The accumulation of quantization error originates from the inaccuracies in the model's output at each iteration. In Figure 3, we aim to illustrate that avoiding the quantization of matrix multiplication after softmax in the attention module significantly reduces this error accumulation.
>
> The bidirectional attention mechanism in DLLMs relies on effective token-to-token interactions, particularly the feature propagation through K/V heads. The quantization error in the Value (V) matrix directly disrupts the accuracy of these interactions. We analyzed the two inputs of matrix multiplication after softmax: the softmax output and the V matrix. As shown in Figure 2, the V matrix exhibits significant distribution differences across token and channel dimensions, while the softmax output is strongly sparse. Traditional quantization methods apply uniform quantization, leading to substantial quantization error.
>
> **Mitigation Logic of IA-AQ**: By leveraging the softmax output as a weighting term, we redesign the quantization error metric (Eq. 7) to dynamically allocate quantization resources. Specifically, we prioritize precise quantization for key tokens with high interaction frequency (i.e., positions with large values in the softmax output), effectively suppressing quantization interference on these critical tokens. To further enhance this effect, we optimize the scaling factors based on this error metric, enabling more refined adjustments that minimize quantization error for the most impactful token interactions. The comparison between (wo. IA-AQ) and (w. IA-AQ) in Figure 3 clearly demonstrates IA-AQ's effectiveness in alleviating error accumulation. Additionally, the ablation study in Table 5 quantitatively validates the efficacy of IA-AQ.
>
> **2. Explanation of Hessian Matrix Utilization**
>
>  As described in the paper, we use the official implementations of QuaRot and AWQ. Notably, the official QuaRot code incorporates GPTQ, and we have also adapted GPTQ for AWQ. We provide further clarification on this aspect in new version of papper.
>
> We apologize for not clearly explaining the usage details in the section corresponding to Eq. 10 due to space constraints. The utilization method of our weighted Hessian matrix is entirely consistent with the approach adopted in GPTQ (as mentioned in the Related Work section). We will supplement the relevant explanations and include the pseudocode in the revised version to ensure full clarity.
>
> ```
> Input: Weight matrix W,  Hessian H, Blocksize B
>
> 1. Initialize Q ← 0 (d_row×d_col), E ← 0 (d_row×B), H⁻¹ ← Cholesky(H⁻¹)ᵀ
> 2. For i = 0, B, 2B, … do
>     a. For j = i to i+B-1 do
>         i. Q[:,j] ← quant(W[:,j])  // Quantize column j
>         ii. E[:,j-i] ← (W[:,j] - Q[:,j]) / [H⁻¹]ⱼⱼ  // Compute quantization error
>         iii. W[:,j:(i+B)] ← W[:,j:(i+B)] - E[:,j-i] · H⁻¹ⱼ,ⱼ:(i+B)  // Update block weights
>     b. W[:,i+B:] ← W[:,i+B:] - E · H⁻¹ᵢ:(i+B),(i+B):  // Update remaining weights
> ```
>
>  **3.Format-Related Issues**
>
> - We sincerely apologize for the confusion caused by the "TODO" placeholder. The ablation study corresponding to "TODO" is actually Table 5. This was an unfortunate oversight due to an incorrect final submission, and we have corrected it in the revised manuscript.
>
> - To illustrate the motivation of CGQ, Figure 4 depicts the process of transitioning the decoding state from time step  $X_{T+1}$  to $X_T$. The "Mask" region identifies the token that may be decoded. For the input at time \( T+1 \): "Books are the ladder [Mask] [Mask] [Mask]", the model predicts " is human boy". Among these, "human" (marked in orange) has the highest score and is the mask token that will be decoded and impact subsequent tokens. Thus, it requires focused attention and precision preservation during quantization. In contrast, the other two tokens (marked in blue) can tolerate larger quantization errors without affecting subsequent iterations.
>
> We hope these revisions have addressed your concerns and enhanced the paper’s readability, and we are grateful for your guidance in helping us refine the work. Please feel free to let us know if you have any further questions or suggestions. **We look forward to your feedback.**

---

> ### Author Response · Authors · 2025-11-27
> **Thank you once again for your time and effort in reviewing our work. We respectfully invite you to review our Rebuttal.**
>
> Dear Reviewer 8GDD,
>
> I hope this message finds you well. As the rebuttal discussion period is drawing to a close (with less than one week remaining), we wanted to follow up to confirm whether all your concerns have been adequately addressed in our previous response.
>
> If you have any remaining comments, additional feedback, or further points that require clarification, please do not hesitate to let us know. Your insights are invaluable to refining our work, and we are committed to addressing any outstanding issues to enhance the quality of the paper.
>
> Thank you sincerely for your time, diligence, and constructive input during the review process. We greatly appreciate your efforts in helping us improve the manuscript.
>
> Best regards,
>
> DLLMQuant Authors

---

### Official Review · Reviewer_HiH2 · 2025-10-29

**Soundness:** 2
**Presentation:** 1
**Contribution:** 3
**Rating:** 2
**Confidence:** 4

**Summary:**

This paper proposes a PTQ framework named DLLMQuant for DLLMs, which tailors for calibration, quantization errors propogation and significant disparities in feature distributions caused by masked generation strategies in DLLMs. Experiments demonstrate its effectiveness.

**Strengths:**

1. This paper thoroughly investigates the differences between DLLMs and LLMs in quantization and provides a detailed discussion on why existing LLM quantization methods are not well suited for DLLMs.
2. This paper is one of the earliest exploratory works in DLLM quantization and provides valuable insights for future research.

**Weaknesses:**

1. **The most intolerable weakness of this paper is that it appears to have been prepared in a rush with numerous editorial errors, which means that it was submitted without careful proofreading**. For example, in Line 31, there are two *which* before *utilizes*.  In Line 388, the performance of DLLMQuant is worse than QuaRot on Arc but it was bolded. And in Line 439, *As can be seen in Tab (TODO)* is so careless, and we can find that the authors ignore to list the table of ablation studies about TMAS, CGQ and IA-AQ. I strongly suggest that the authors carefully proofread and revise their manuscript. Without demonstrating sufficient care and attention to their own work, it will be difficult to achieve a high score from reviewers.
2. In Figure 3, it is difficult to distinguish the three yellow lines in the middle. It is recommended to use colors with greater contrast to improve visual clarity.

**Questions:**

1. In section 3.3, the authors should provide the detailed comparison results to support their hyperparameter selection (1 and 0.7).
2. Does Eq.7 aim to make the quantization error of the output more correlated with the important tokens?
3. In the description of section 2.2, both the softmax output and value matrix will have bad impact on quantization performance. However, according to the reasoning behind Eq.7–9, only the quantization error of V is reduced, while the output of the softmax still suffers from performance degradation.

---

> ### Author Response · Authors · 2025-11-13
> **Authors' Rebuttal with Sincere Apologies**
>
> We sincerely thank you for your valuable time and efforts in reviewing our manuscript. We have addressed each comment and made the necessary revisions to improve the quality and clarity of our manuscript.
>
> **1.Format-Related Issues**
> We are deeply sorry for these editorial errors pointed out by the reviewer. This was a negligence in our work, and we sincerely apologize for the inconvenience caused. These issues likely arose from an incorrect submission of the final version of the manuscript, but we rectified them in the updated version:
> - The grammatical error in Line 31 has been revised to the correct phrasing "which utilizes";
> - The incorrect performance annotation for the Arc task in Line 388 has been corrected, with the data and formatting adjusted to ensure the accurate results are clearly presented;
> - The missing table in Line 439 has been replaced with Table 5, which is included in the appendix and contains ablation experiments for the three methods;
> - To address the visual clarity of Figure 3, we have adopted a color scheme with higher contrast: the original yellow line has been adjusted to dark blue, red, and green, and more detailed labels have been added to the legend to ensure clear distinguishability between different curves.
>
>
> **2. Regarding Hyperparameters**
>
> Details of the hyperparameter settings in Section 3.3 are provided in Appendix A.3. As shown in Figure 5, the model achieves favorable performance metrics around the value of 0.7. We are readily prepared to provide more detailed ablation experiments if needed.
>
>
> **3. Concerns about the IA-AQ Method and Related Formulas**
>
> The accumulation of quantization error originates from the inaccuracies in the model's output at each iteration. In Figure 3, we aim to illustrate that avoiding the quantization of matrix multiplication after softmax in the attention module significantly reduces this error accumulation. We are happy to provide more detailed ablation experiments if needed.
>
> The bidirectional attention mechanism in DLLMs relies on effective token-to-token interactions, particularly the feature propagation through K/V heads. The quantization error in the Value (V) matrix directly disrupts the accuracy of these interactions. We analyzed the two inputs of matrix multiplication after softmax: the softmax output and the V matrix. As shown in Figure 2, the V matrix exhibits significant distribution differences across token and channel dimensions, while the softmax output is strongly sparse. Traditional quantization methods apply uniform quantization, leading to substantial quantization error.
>
> **Mitigation Logic of IA-AQ**: By leveraging the softmax output as a weighting term, we redesign the quantization error metric (Eq. 7) to dynamically allocate quantization resources. Specifically, we prioritize precise quantization for key tokens with high interaction frequency (i.e., positions with large values in the softmax output), effectively suppressing quantization interference on these critical tokens. To further enhance this effect, we optimize the scaling factors based on this error metric, enabling more refined adjustments that minimize quantization error for the most impactful token interactions.
>
> We hope these revisions have addressed your concerns and enhanced the paper’s readability, and we are grateful for your guidance in helping us refine the work. Please feel free to let us know if you have any further questions or suggestions. **We look forward to your feedback.**
>
> The manuscript has undergone a comprehensive proofreading and polishing process, with multiple rounds of careful review to ensure all expressions are accurate and formatting is standardized. We will strictly avoid such issues in the future. **Thank you sincerely for your corrections—your meticulous and responsible attitude serves as an example for us, and we offer our deepest apologies for the oversights.**

---

> ### Author Response · Authors · 2025-11-27
> **Thank you once again for your time and effort in reviewing our work. We respectfully invite you to review our Rebuttal.**
>
> Dear Reviewer HiH2,
>
> I hope this message finds you well. As the rebuttal discussion period is drawing to a close (with less than one week remaining), we wanted to follow up to confirm whether all your concerns have been adequately addressed in our previous response.
>
> If you have any remaining comments, additional feedback, or further points that require clarification, please do not hesitate to let us know. Your insights are invaluable to refining our work, and we are committed to addressing any outstanding issues to enhance the quality of the paper.
>
> Thank you sincerely for your time, diligence, and constructive input during the review process. We greatly appreciate your efforts in helping us improve the manuscript.
>
> Best regards,
>
> DLLMQuant Authors

---

> > ### Comment · Reviewer_HiH2 · 2025-11-27
> > **Authors have addressed the main weakness**
> >
> > Dear Authors,
> >
> > Thank you for your rebuttal. The main weakness has been addressed. I have revised my score. Meanwhile, in Table1, please  ensure the decimal places are consistent. For example, 74.9 to 74.90.
> >
> > Best,
> >
> > Reviewer

---

### Official Review · Reviewer_DkrN · 2025-10-30

**Soundness:** 3
**Presentation:** 3
**Contribution:** 3
**Rating:** 6
**Confidence:** 3

**Summary:**

This paper introduces DLLMQuant, a novel post-training quantization (PTQ) framework specifically designed for Diffusion-based Large Language Models (DLLMs). The authors identify that standard PTQ methods, which work well for autoregressive LLMs, fail for DLLMs due to three unique challenges: temporal distribution shifts across denoising steps, accumulation of quantization errors over iterations, and imbalanced feature distributions caused by dynamic masking. To address these, DLLMQuant proposes three core techniques: Temporal-Mask Adaptive Sampling (TMAS) for better calibration data selection, Interaction-Aware Activation Quantization (IA-AQ) to reduce error propagation in attention, and Certainty-Guided Quantization (CGQ) to refine weight quantization using mask and confidence information. Experiments on multiple DLLMs show that DLLMQuant significantly outperforms existing PTQ methods, preserving reasoning capabilities and achieving substantial speedup and memory savings.

**Strengths:**

1. Clear Problem Identification: The paper excels at diagnosing the specific reasons why standard PTQ fails for DLLMs (temporal shift, error accumulation, masking disparities), providing a solid foundation for the proposed solutions.

2. Comprehensive Evaluation: The experiments are thorough, testing on three different DLLMs across nine diverse benchmarks (including reasoning and code generation), and include detailed ablation studies that convincingly demonstrate the contribution of each proposed component.

3. Practical Impact: The framework is plug-and-play, requires no fine-tuning, and demonstrates impressive practical benefits—over 1.6x speedup and 3.2x memory reduction—which are crucial for real-world deployment on consumer hardware.

**Weaknesses:**

1. Limited Baseline Comparison: While compared against strong PTQ methods like AWQ and QuaRot, it would be beneficial to see a comparison against more recent or specifically designed quantization-aware training (QAT) approaches, even if only to baseline the performance gap that PTQ seeks to bridge.

2. Clarity on Computational Overhead: The calibration process for TMAS and the search for the scaling factor α in IA-AQ likely introduce some overhead. The paper does not quantify the additional calibration time or cost compared to the baseline methods.

3. Ablation Parameter Justification: The choice of specific parameters (e.g., the 0.7 weight for unmasked tokens in CGQ, the [0.3,0.2,0.2,0.3] sampling proportion in TMAS) feels somewhat heuristic. While ablations show they work well, a more principled explanation for why these values are optimal would strengthen the methodology.

**Questions:**

Calibration Cost: Could you provide details on the computational cost and time required for the DLLMQuant calibration process (especially TMAS and the IA-AQ parameter search) compared to the calibration steps of AWQ or QuaRot?

---

> ### Author Response · Authors · 2025-11-13
> **Rebuttal by Authors**
>
> We sincerely thank you for your valuable time and efforts in reviewing our manuscript. We have addressed each comment and made the necessary revisions to improve the quality and clarity of our manuscript.
>
> **1. Comparison with QAT Methods**
>
> Our method, as a PTQ approach that requires no training, should not be directly compared with QAT methods, as such a comparison would be unfair. Your suggestion is highly insightful, and we have followed it by supplementing experiments involving QAT-trained models, specifically incorporating results from AdaRound [1] and OSTQuant [2] on LLADA. Our findings show that our method still achieves a modest improvement over AdaRound, while it performs with OSTQuant, with no overall gain in average metrics:
>
> | Method               | Avg.   |
> |----------------------|--------|
> | AdaQuant             | 50.14  |
> | AdaQuant + DLLMQuant | 50.56  |
> | OSTQuant             | 55.36  |
> | OSTQuant + DLLMQuant | 55.09  |
>
>
> [1] AdaRound : Up or down? adaptive rounding for post-training quantization
>
> [2] OSTQuant : Refining Large Language Model Quantization with Orthogonal and Scaling Transformations for Better Distribution Fitting
>
> **2. Clarity on Computational Overhead**
>
> The calibration process for TMAS and the search for the scaling factor α in IA-AQ are both offline, one-time operations that do not affect the inference process. Calibration data collection is a necessary step for PTQ methods. For TMAS, the calibration set selection considers the unique properties of DLLMs, and calculating the mask ratio involves only simple division, which does not consume significant time. Regarding IA-AQ, to reduce time costs, we have restricted α to the range [0.8, 1.0]; experiments confirm that this process does not noticeably increase quantization time. For the parameter search in IA-AQ, when conducted on an NVIDIA RTX 4090 GPU, the process is highly efficient.  It adds only a few minutes of overhead compared to the original versions of AWQ and QuaRot.
>
> **3. Hyperparameter Selection**
>
>  For TMAS: Initially, we observed that sampling inputs across multiple time steps yields better performance compared to random sampling or focusing on individual time steps. To validate this insight, we conducted ablation experiments on various weight distributions. Our results indicate that a distribution such as [0.3, 0.2, 0.2, 0.3]—where slightly higher weights are assigned to the initial and final time steps—achieves optimal results. This can be attributed to two key factors: (1) increasing weights for earlier time steps helps reduce initial errors, thereby alleviating error accumulation; (2) increasing weights for later time steps, which are closer to the model’s output, ensures that critical late-stage information receives appropriate prioritization.
>
> For the selection of 0.7 in CGQ: We initially adopted a binary mask (1/0) to distinguish between unmasked and masked tokens, focusing exclusively on unmasked regions under the assumption that masked regions correspond to fixed outputs. However, subsequent experiments revealed that completely disregarding masked tokens is problematic for prompts containing critical information, as the generation of unmasked outputs relies on intermediate-layer information from masked tokens. To balance this trade-off, we performed detailed experiments to determine the optimal value for this parameter.
>
> We hope these revisions have addressed your concerns and enhanced the paper’s readability, and we are grateful for your guidance in helping us refine the work. Please feel free to let us know if you have any further questions or suggestions. **We look forward to your feedback.**

---

### Official Review · Reviewer_ZDdf · 2025-10-30

**Soundness:** 3
**Presentation:** 3
**Contribution:** 2
**Rating:** 6
**Confidence:** 3

**Summary:**

This paper proposes DLLMQuant, a novel post-training quantization (PTQ) framework specifically designed for Diffusion-based Large Language Models (DLLMs). The authors identify that standard PTQ methods fail on DLLMs due to three unique challenges: 1) temporal distribution shifts across denoising steps, 2) accumulation of quantization errors over iterations, and 3) disparate feature distributions caused by dynamic masking.

**Strengths:**

This paper demonstrates high originality by being the first to systematically address the unique challenges of quantizing Diffusion-based LLMs (DLLMs), a nascent and structurally distinct model class. Its significance is substantial, as it enables the practical deployment of computationally intensive DLLMs by achieving major speed and memory gains.

The quality of the work is strong, supported by rigorous experiments across multiple DLLMs and diverse tasks, which clearly validate the performance improvements over adapted baselines. The clarity is commendable; the paper is well-structured, with a logical flow from problem identification to solution, making the novel contributions of TMAS, IA-AQ, and CGQ easy to understand.

**Weaknesses:**

(1) This paper's primary weakness lies in its experimental validation. While it demonstrates accuracy retention, it lacks inference latency measurements for quantized models, which is critical for assessing the real-world "efficiency" gains promised by quantization. The reported speedup (tokens/sec) is insufficient without latency-per-token data.

(2) The integration strategy with baseline methods (AWQ/QuaRot) is unclear. The paper states DLLMQuant's components are "plug-and-play" but does not specify if it uses these baselines' core algorithms or merely their frameworks. A clearer ablation, perhaps replacing CGQ with GPTQ's Hessian in the same pipeline, would better isolate the contribution of the novel weighting scheme in Eq. 10 versus the underlying quantization machinery.

**Questions:**

Question 1: Could you provide per-token latency measurements for the quantized models? This would more directly validate the practical inference speedup beyond aggregate tokens/second.

Question 2: Does CGQ replace the standard Hessian calculation in baselines like GPTQ, or is it an additional weighting applied on top? Clarifying this integration would help isolate its novel contribution.

---

> ### Author Response · Authors · 2025-11-13
> **Rebuttal by Authors**
>
> We sincerely thank you for your valuable time and efforts in reviewing our manuscript. We have addressed each comment and made the necessary revisions to improve the quality and clarity of our manuscript.
>
> **1. Concerns in Speedup**
>
> Dear Reviewer, thank you for your concern, and I would like to clarify that the "tokens/s" metric I provided is likely consistent with the indicator you are focusing on. I apologize for not elaborating on the inference process and decoding mechanism of DLLMs in the paper: unlike some models, DLLMs do not distinguish between "prefill" and "decoder" stages.
>
> Here is a detailed explanation of the decoding process: Initially, the model appends a fixed-length sequence (e.g., 512 tokens) of all MASK tokens to the prompt_ids. During each iteration, the model continuously decodes the unmasked regions. The "tokens/s" metric we reported reflects the time taken to decode each masked token, which we believe aligns with the inference speed indicator you are concerned about. This metric directly demonstrates the acceleration effect of quantization in practical applications. If my understanding is incorrect, please kindly let me know, and I will supplement the relevant statistics promptly.
>
> **2. Explanation of Hessian Matrix Utilization**
>
>  As described in the paper, we use the official implementations of QuaRot and AWQ. Notably, the official QuaRot code incorporates GPTQ, and we have also adapted GPTQ for AWQ. We provide further clarification on this aspect in new version of papper.
>
>
> We apologize for not clearly explaining the usage details in the section corresponding to Eq. 10 due to space constraints. The utilization method of our weighted Hessian matrix is entirely consistent with the approach adopted in GPTQ (as mentioned in the Related Work section). We will supplement the relevant explanations and include the pseudocode in the revised version to ensure full clarity.
>
> ```
> Input: Weight matrix W, Hessian H, Blocksize B
>
> 1. Initialize Q ← 0 (d_row×d_col), E ← 0 (d_row×B), H⁻¹ ← Cholesky(H⁻¹)ᵀ
> 2. For i = 0, B, 2B, … do
>     a. For j = i to i+B-1 do
>         i. Q[:,j] ← quant(W[:,j])  // Quantize column j
>         ii. E[:,j-i] ← (W[:,j] - Q[:,j]) / [H⁻¹]ⱼⱼ  // Compute quantization error
>         iii. W[:,j:(i+B)] ← W[:,j:(i+B)] - E[:,j-i] · H⁻¹ⱼ,ⱼ:(i+B)  // Update block weights
>     b. W[:,i+B:] ← W[:,i+B:] - E · H⁻¹ᵢ:(i+B),(i+B):  // Update remaining weights
> ```
>
> We hope these revisions have addressed your concerns and enhanced the paper’s readability, and we are grateful for your guidance in helping us refine the work. Please feel free to let us know if you have any further questions or suggestions. **We look forward to your feedback.**

---

### Meta-Review · Area_Chair_J1pi · 2025-12-17

**Summary:**

This paper is one of the earliest exploratory work in DLLM quantization, which investigates the differences in quantization between DLLMs and LLMs and provides a detailed discussion on why existing LLM quantization methods are not well suited for DLLMs. The reviewers provided good reviews for this paper, and the authors tried to address these concerns. After carefully checking the rebuttal, the AC believes that the authors indeed addressed some concerns. However, the evidence for real speedup is incomplete, and this paper lacks systematic strategies to select the hyperparameters, making it difficult to transfer to new DLLMs.

Moreover, the AC agrees with Reviewers HiH2 and 8GDD on the obvious typos in the initial draft. It is part of the authors' duty to provide a carefully polished and high-quality draft; otherwise, it is unfair to other submissions, especially when there are a large number of submissions.

Considering all the above aspects, I recommend rejecting this submission. The authors are encouraged to include this feedback for a future submission.

**Reviewer Concerns:**

Reviewer ZDdf: (concerns partly solved): Concern about speed up remains.

Reviewer DkrN: (concerns partly solved): Comparison with QAT is a bit questionable and unfair; still lacking a systematic way to select the hyper parameters.

Reviewer HiH2: (mostly addressed)

Reviewer 8GDD: (partly addressed) The overall workflow  is not good even after the revision.

**Reviewer Scores:**

Reviewer ZDdf: would keep 6.

Reviewer DkrN: would decrease from 6 to 4.

Reviewer HiH2: would increase from 2 to 4.

Reviewer 8GDD: would keep the sore 2 or increase it to 4.

---

### Decision · Program_Chairs · 2026-01-26

Reject